# Rat Hepatocytes Mitigate Cadmium Toxicity by Forming Annular Gap Junctions and Degrading Them via Endosome–Lysosome Pathway

**DOI:** 10.3390/ijms232415607

**Published:** 2022-12-09

**Authors:** Junzhao Yuan, Xiaoqian Huang, Yumeng Zhao, Jianhong Gu, Yan Yuan, Zongping Liu, Hui Zou, Jianchun Bian

**Affiliations:** 1College of Veterinary Medicine, Yangzhou University, 12 Wenhui East Road, Yangzhou 225009, China; 2Jiangsu Co-Innovation Center for Prevention and Control of Important Animal Infectious Diseases and Zoonoses, Yangzhou 225009, China; 3Joint International Research Laboratory of Agriculture and Agri-Product Safety of the Ministry of Education of China, Yangzhou University, Yangzhou 225009, China

**Keywords:** annular gap junction, cadmium, connexin 43, endosome, lysosome, microtubule

## Abstract

Gap junction protein connexin 43 (Cx43) plays a critical role in gap junction communication in rat hepatocytes. However, those located between hepatocytes are easily internalized following exposure to poisons. Herein, we investigated the potential of buffalo rat liver 3A (BRL 3A) cells to generate annular gap junctions (AGJs) proficient at alleviating cadmium (Cd) cytotoxic injury through degradation via an endosome–lysosome pathway. Our results showed that Cd-induced damage of liver microtubules promoted Cx43 internalization and increased Cx43 phosphorylation at Ser373 site. Furthermore, we established that Cd induced AGJs generation in BRL 3A cells, and AGJs were subsequently degraded through the endosome–lysosome pathway. Overall, our results suggested that Cx43 internalization and the generation of AGJs were cellular protective mechanisms to alleviate Cd toxicity in rat hepatocytes.

## 1. Introduction

Cadmium (Cd), a kind of heavy metal element, is discharged into the environment through industrial waste from mining or the manufacture of electronics [1,2,3]. Cd element mainly exists in nature as greenockite (CdS). Cd element in the form of the elementary substance is non-toxic. The ion form of Cd^2+^ from mining activities and industrial wastes is harmful, it can cause oxidative stress damage and induce cancers in animals and humans [4]. Hence, aquatic or terrestrial organisms in Cd-polluted areas are exposed to Cd toxicity [5,6,7]. After animals ingest Cd^2+^, it is absorbed through the digestive system, accumulated and stored in the liver, and gradually transported to the kidney and other organs through blood circulation. After Cd^2+^ enters cells, it will extensively combine with metallothionein, phospholipids, nucleic acids, and other biological macromolecules in cells, resulting in damage to organelles such as mitochondria, endoplasmic reticulum, and lysosome, and eventually induce cells to undergo apoptosis [8,9,10]. More seriously, Cd in animals is easily accumulated in the bones, and its half-life is as long as 30 years, which continues to affect the health of animals [11].

Connexins play important roles in the cellular communication of various tissues and the occurrence or regulation of various diseases such as hearing, cardiovascular, cutaneous, and neurological diseases [12,13,14]. Gap junctions composed of connexins are essential for survival and intercellular communication between hepatocytes [15]. Connexins form a half-channel (connexon) composed of hexamers and interlink with the half-channels on adjacent cell membranes to form an integral gap junction channel. Hundreds to thousands of channels locally aggregate in the cell membrane and dock with the channels on adjacent cells to form the so-called gap junction plaques (GJPs) [16,17]. Connexins are rapidly and continuously generated, internalized, degraded, or recycled to adapt to cellular environmental changes, such as cell growth, migration, and stress [18]. Under physiological conditions, minimal levels of connexins are internalized and regenerated [19]; however, they undergo extensive internalization and degradation under certain stimuli. Connexins can form annular gap junctions (AGJs) with double membranes through intracellular invagination under the influence of various factors, which are endocytosed by the cells in vitro [16]. Connexin 43 (Cx43) has been widely studied because of its universal distribution between cells. The normal and abnormal expression patterns of Cx43 in myocardial, vascular, nerve, and tumor cells have always been a research hotspot [20,21,22,23,24]. However, the regulatory mechanism of Cx43 degradation, especially the internalization and degradation stimulated by different poisons, is yet to be fully elucidated.

The conversion of connexins under physiological conditions is achieved by endocytic small endocytic double-membrane vesicles [25]. Phosphorylation and ubiquitination at the C-terminal of Cx43 jointly regulate the initiation of Cx43 internalization [18,26,27,28]. Phosphorylation at Ser373, Ser279/282, Ser262, and Ser368 sites on Cx43 has been extensively studied and all of them promote the internalization of Cx43 degradation; specifically, increased phosphorylation at Ser373 has been shown to attenuate the interaction between Cx43 and the tight junction protein zonula occludens-1 (ZO-1), in turn promoting the internalization of GJP [27].

The cytoskeleton consists of microtubules, microfilaments, and intermediate filaments, with microtubules being distributed within cells originating around the nucleus and densely distributed in the cytoplasm. They play an important role in the movement and localization of mitochondria, lysosomes, and other organelles. Endosomes are single-layered vesicles derived from the endoplasmic reticulum or Golgi apparatus [29,30,31]. Endosomes are also referred to as secondary lysosomes because of their gradual acidification and eventual fusion with lysosomes through the lysosomal progenitor cells of the endoplasmic reticulum during maturation [32,33]. Endosomes depend on the cytoskeleton to undergo maturation from early endosomes (EE; signature protein: EEA1) to multivesicular bodies (MVBs) and late endosomes (LE, signature protein: Rab7) [30,34,35]. AGJs internalized between cells can fuse with endosomes; thus, the connexins on the inner membrane will be broken into smaller AGJs and transported to lysosomes for degradation. In contrast, the connexins on the outer membrane may be transported back to the cell membrane for recycling or lysosomal degradation [36,37].

Whether the heavy metal Cd can induce the production of annular gap junctions in hepatocytes is unknown, and our study mainly relied on the immunofluorescence technique to explore this possible outcome. Our study findings will provide an *in vitro* perspective of the internalization of connexins in inhibiting Cd-induced cell damage and a rare intercellular phenomenon.

## 2. Results

### 2.1. Cd-Induced Microtubule Damage Promotes Cx43 Internalization and Degradation Accompanied by Increased Phosphorylation at Ser373 Site

Our findings showed that as Cd concentration increased, Cx43 gradually accumulated from its position in the cell membrane where the microfilament structure was located to the cell interior and was accompanied by a significant decrease in the fluorescence intensity corresponding to Cx43 (Figure 1A). Similarly, BRL 3A cells were treated with 0, 5, 10, and 20 μM Cd for 12 h, followed by immunofluorescence with anti-α-tubulin-FITC antibody and anti-Cx43 antibody co-staining. Following this, microtubules became sparse and disordered with increasing Cd concentration (Figure 1B). However, Cx43 marker sites gradually distributed to the interior of the cell, and co-localization with microtubules gradually increased. In some regions, Cx43 formed large clusters beside the cell membrane. Nevertheless, the Western blotting analysis showed that Cx43 levels significantly decreased upon treatment with Cd at 10µM and 20 µM (Figure 1C,D). Moreover, Western blotting analysis showed that the level of P-Cx43 increased proportionally with Cd concentration and reached its highest level at 10 μM Cd (Figure 1E,F). 

### 2.2. Cd-Induced Damage to Microtubules Promotes Cx43 Internalization

Intracellular distribution of Cx43 after treatment of hepatocytes with Cd or nocodazole was observed using immunofluorescence. The images showed that Cd significantly reduced the intracellular distribution and fluorescence intensity of Cx43. Nocodazole alone or in combination with Cd caused the decrease in Cx43 in hepatocytes, leading to a decreased intensity in Cx43 fluorescence (Figure 2A). Subsequently, Western blotting analysis was used to determine the effect of nocodazole on Cx43 degradation. The results showed that combination treatment with Cd and nocodazole decreased Cx43 levels, suggesting that destabilizing the microtubules promotes internalized degradation of Cx43 (Figure 2B,C). In addition, nocodazole alone or in combination with Cd increased the level of P-CX43 (Ser373) when compared to the control and Cd group, suggesting that damaged microtubules were also an important factor in the internalization of Cx43 (Figure 2D,E). Therefore, our study findings showed that microtubule damage from Cd toxicity promotes the internalization of Cx43 and, to some extent, promoted the degradation process of Cx43.

### 2.3. Cd Exposure Induces the Formation of AGJs in Rat Hepatocytes

The results described above confirmed that the optimal internalization of Cx43 was achieved when hepatocytes were treated with 10 μM Cd for 12 h (Figure 2D,E). The discovery of AGJs in BRL 3A cells was accidental since the internalization of adjacent cell boundaries to form an annular gap junction is an uncommon observation (Figure 3A). These AGJs could also be observed using a transmission electron microscope, and most regions depict extensive Cx43 internalization activity (Figure 3B). We measured the diameters of all AGJs in the visual field of all Cd groups and compared them with the diameters of normal Cx43 spots in control group. The results showed that the diameters of AGJs induced by Cd were approximately three times those of Cx43 spots in the control group (Figure 3C). We could observe only one or two distinct, large, regularly structured AGJs within a group of cells. Most AGJs were distributed in the cell-cell junctions (Figure 3Db); however, some AGJs were also distributed on the other side of the cells, away from cell junctions (Figure 3Dc). Amazingly, some AGJs were observed outside the cell (Figure 3Dd). 

### 2.4. Cd Treatment Caused the Degradation of Internalized AGJ through Endosome-Lysosome Pathway

We found that Cd could induce the trafficking of internalized AGJ through early/late endosomes and eventually fuse with lysosomes. To determine whether the internalized AGJs could be degraded by the endosome–lysosomal pathway, the anti-Cx43 antibody was co-stained with early endosomal (Figure 4A), late endosomal (Figure 4B), and lysosomal markers (Figure 4C) with anti-EEA1 antibody, anti-Rab7a antibody, and anti-LAMP2 antibody, respectively. The results showed that Cx43 could be co-localized with EEA1, Rab7a and LAMP2, which demonstrated that Cx43 could be degraded by the endosome–lysosome pathway.

### 2.5. Internalization of Cx43 Protects Rat Hepatocytes from Cd-Induced Apoptosis and Alleviates Cell Damage 

TPA is a PKC activator that phosphorylates Cx43 at Ser368 site to promote Cx43 degradation [36]. Herein, TPA was used to internalize Cx43 on the cell membrane. Rat hepatocytes were treated with Cd (10 μM) and TPA (0.5 μM in DMSO) alone or in combination for 12 h. Compared with control and Cd groups, cell morphology observed under a light microscope suggested that co-treatment with TPA and Cd significantly reduced cell death (Figure 5A). Next, Hoechst 33258 was used to stain damaged nuclei to assess Cd toxicity in rat hepatocytes and the role of TPA in mitigating Cd toxicity. The results showed that compared with the Cd group, the combined treatment of rat cells with TPA and Cd alleviated the damage to the nucleus (Figure 5B). Similarly, CCK8 assay results also showed that the combined treatment with TPA and Cd reduced the damage caused by Cd toward cell activity (Figure 5C). The levels of Cx43 and three apoptotic protein markers, activated-caspase 3, Bax, and Bcl-2, were determined using Western blotting analysis. The results showed that the co-treatment of hepatocytes with TPA and Cd significantly decreased levels of Cx43 and inhibited Cd-induced apoptosis (Figure 5D–G). Through transmission electron microscopy, it was observed that the nucleus deformation was reduced in the Cd group, and more lysosomal structures appeared in cells, while the combination of Cd and TPA resulted in more lysosomal organelles in cells but alleviated the nuclear deformation and damage (Figure 5H). These results suggest that the internalization of Cx43 may be an active protective mechanism in rat hepatocytes.

## 3. Discussion

In this study, we observed that the heavy metal Cd induced AGJ production in rat hepatocytes *in vitro*. Various environmental toxins can induce the internalization and degradation of gap junction proteins [38,39]. However, to date, researchers have only observed the internalization and degradation of AGJ caused by biological factors under in vitro conditions [40,41]. We found that the large AGJs (diameter up to 3–4 μm) can be ten times the size of normal small-sized AGJs (about 1μm in diameter). In addition, we found that the increase in Cd concentration caused noticeable damage to the microtubular structure in BRL 3A cells and Cd-induced microtubule damage participated in promoting the internalization and degradation of Cx43. The microtubule inhibitor, nocodazole, can cause elevated phosphorylation levels of Ser373-Cx43 associated with the initiation of internalization of Cx43.

Studies have shown that the GJP between two adjacent cells can interact with microfilaments, microtubules, and tight junction proteins such as ZO-1 and β-catenin, these diverse structural proteins aggregate to form stable submembrane protein complexes that allow intercellular communication and energy transfer between cells [42,43]. Acute Cd exposure can disrupt the polymerization of cytoskeletal protein monomers and depolymerize microtubules [44,45]. Herein, we found that the increase in Cd concentration caused noticeable damage to the microtubules, which are critical for the local stability of cell adhesion structures, such as kinds of connexins and tight junctions. Cd promoted an increase in Cx43 phosphorylation at the Ser373 site. After co-treatment with nocodazole and Cd to depolymerize the microtubules, phosphorylation levels of Cx43 at Ser373 increased significantly, providing evidence that microtubules play an important role in inducing Cx43 internalization.

Interestingly, since hepatocytes grow in a tightly connected manner, most of the AGJs induced by Cd were present at cell-cell junctions. However, some AGJs were present away from cellular junctions. There were no apparent traces of cell contact in these regions; therefore, these AGJs were suspected to be derived from a single-cell monolayer membrane. The other possibility is that these AGJs were derived from the internalized AGJs transported from one side of the cell to the other along the microtubule. Another interesting phenomenon is that AGJs can also be found outside cells, implying that hepatocytes would have exocytosed them. It has been reported that Cx43 can be found on the exosome membrane because the connexins on this membrane can provide a bridge for the intercellular exchange between exosomes and host cells [46,47,48]. Since internalized AGJs are via the endosome–lysosome pathway or the autophagy-lysosome pathway, study has shown that Cd can inhibit autophagic flux by affecting lysosomal acidity or destroying autophagy–lysosome fusion to activate the exosome or secretory autophagy pathways [49], we speculate that the AGJs that cannot be effectively degraded are discharged outside the cell via phagocytosis.

Cx43 internalization is related to complex ubiquitination or phosphorylation modification at the C-terminus of Cx43 [27,50]. The phosphorylation at the Ser 373 site can inhibit the binding between Cx43 and ZO-1, thereby controlling the size of GJP. The mutation of the Cx43 sequence affecting the Ser373 site can reduce the probability of Cx43 being phosphorylated and thus affect the normal binding of ZO-1 with Cx43, weakening the internalization of GJP, thereby causing it to remain stable for a longer period [27]. In our study, the destruction of microtubules from Cd toxicity or microtubule inhibitor nocodazole caused an increase in phosphorylation at Ser373. The destruction of microtubules may have an indirect rather than direct effect on the increase in Ser373 phosphorylation because microtubule damage affects a variety of organelles and thus causes drastic changes in the intracellular environment, which may in turn affect the activity of corresponding kinases. Nevertheless, compared to the treatment with Cd alone, the combined treatment with nocodazole and Cd destroyed microtubules and caused a significant decrease in Cx43, indicating that the damage to microtubules by Cd destabilizing microtubules promotes internalized degradation of Cx43. 

Endosomes are important organelles for the effective processing of membrane proteins after internalization. The endosomal sorting complexes required for transport (ESCRT) molecular machinery on early endosomes can ubiquitinate or phosphorylate proteins such as connexins on the endosomal membrane according to cellular requirements to determine the ultimate fate of these membrane proteins [51]. They are either transported to the cell membrane for recycling or the lysosome for degradation. Leithe et al. found that when TPA was used to treat IAR20 rat hepatocyte cells, the internalized AGJs were degraded by the endosome–lysosome pathway [36]. Through immunofluorescence staining, we found that, compared to the control group, Cx43 co-localized with the early endosomal marker EEA1, the late endosomal marker Rab7A, and the lysosomal marker LAMP2. Collectively, our results proved that the endosome–lysosome pathway plays a role in Cd-induced internalization and degradation of Cx43.

The Cd-induced decrease in Cx43 levels in BRL 3A cells as well as the internalization and degradation of Cx43 may have blocked the transmission of toxins between cells through gap junctions. Previously, other members of our group used 18β-glycyrrhetinic acid, a compound that promoted Cx43 internalization, to alleviate the damage caused by Cd exposure to rat hepatocytes [52]. The results from the present study are concordant with their results. Under an optical microscope, we observed that the combined treatment of rat hepatocytes with TPA and Cd significantly reduced Cd-induced cell death. In addition, CCK8 toxicity results proved that this combined treatment significantly reduced cytotoxicity compared to Cd monotherapy. Furthermore, we established through immunoblotting experiments and staining of cell nuclei with Hoechst 33258 that co-treatment with TPA and Cd reduced Cd-induced apoptosis, indicating that the internalization of Cx43 is a self-protection mechanism against Cd-induced damage. Therefore, in this study, we confirmed that Cd induces Cx43 internalization in BRL 3A cells, a protective mechanism which cells use to prevent Cd toxicity. Although our experiments demonstrated that Cx43 internalization within 12 h was protective against Cd toxicity, intense Cx43 internalization, especially when cells formed AGJ to internalize and degrade Cx43 on a large scale, was temporary in saving Cd-induced single-cell death. However, to the extent that cell populations can reduce the amount of cell death that can cause damage to more cells by decreasing communication between cells, it may be an effective way to survive within liver tissue.

## 4. Materials and Methods

### 4.1. Materials

Fetal bovine serum was obtained from Solarbio Company (Beijing, China), and cadmium chloride (CdCl2, purity of ≥ 99.99%, USA) was obtained from Sigma company (St. Louis, MO, USA). Phorbol 12-myristate 13-acetate (TPA, P167764) and nocodazole (N129755) were obtained from Aladdin company (Shanghai, China), and Hochest 33258 (23491-44-3) was obtained from MedChemExpress company (New Jersey, USA). Anti-α-tubulin-FITC antibody and anti-LAMP2 antibody were obtained from Sigma, the monoclonal anti-connexin 43 antibody (CX-1B1, 13-8300) was obtained from Invitrogen company (Waltham, MA, USA), anti-EEA1 antibody, anti-bax (ab32503) antibody, and anti-bcl-2 (ab196495) antibody were obtained from Abcam company (Cambridge, UK), anti-RAB7A antibody was obtained from ABclonal company (Wuhan, China), anti-phospho-connexin 43/GJA1 (Ser373; AF8264) was obtained from Affinity Biosciences (Cincinnati, OH, USA), and mouse anti-active caspase-3 (bsm-33199M) was obtained from BIOSS company (Beijing, China). 

### 4.2. Cell Culture and Morphological Observation

BRL 3A cells were purchased from the Cell Bank of the Shanghai Institutes for Biological Sciences, Chinese Academy of Sciences. Cells were cultured in Dulbecco’s Modified Eagle’s Medium (DMEM) supplemented with 10% fetal bovine serum. When cells attained 50% confluence, they were treated with Cd for 12 h. The morphology of hepatocytes was observed under as Leica inverted fluorescence microscope (Wetzlar, Germany), and bright-field photographs were obtained. Optimal concentrations of Cd and nocodazole (an antitumor drug that works by depolymerizing microtubules, 31.25 nM in DMSO) were selected based on our previous work [53]. The optimal concentration of TPA was selected based on the cell counting kit 8 (CCK8) cytotoxicity assay results (Appendix A).

### 4.3. Immunofluorescence Assay

BRL 3A cells were seeded in a 24-well plate at a density of 1 × 10^4^ cells per well and cultured in an incubator. When cells attained 50 confluences, the following independent immunofluorescence staining experiments were carried out:Different concentrations of Cd (0, 5, 10, and 20 µM) were added to 24-well plates to co-stain Cx43 and microfilaments.Different concentrations of Cd (0, 5, 10, and 20 µM) were added to 24-well plates to co-stain Cx43 and α-tubulin.The control, Cd, nocodazole, and nocodazole + Cd were added to 24-well plates. Cells were then stained for Cx43.Three experimental groups were set up in 24-well plates. The control and Cd were added to two wells in each group, respectively. Cells of the three groups were then stained with antibodies from different species against Cx43 and EEA1, Cx43 and Rab7a, Cx43, and LAMP2.The control, Cd, TPA (0.5 µM in DMSO), and Cd + TPA were added to 24-well plates. The cells were then stained with Hochest 33258.

After the above-mentioned procedures, cells were cultured at 37 °C for 12 h in an incubator, the medium was discarded, and cells were washed thrice with ice-cold phosphate-buffered saline (PBS) solution. Cells were fixed with 4% paraformaldehyde solution (0.5 mL) and incubated at 4 °C for 30 min. 0.1% Triton X-100 solution (Beyotime, Shanghai, China) was added to the cells for 2 min to permeabilize the cell membrane. After washing, 0.5% BSA solution (0.5 mL) was added to each well prior to incubation for 30 min at room temperature. The following primary antibodies were individually added to the wells and incubated overnight: anti-α-tubulin-FITC antibody (1:500), anti-connexin 43 antibody (1:200), anti-EEA1 antibody (1:200), anti-Rab7 antibody (1:200), and anti-LAMP2 antibody (1:200). The primary antibody, anti-α-tubulin-FITC antibody, was collected directly, then nuclei were stained with DAPI, cells were mounted. Anti-EEA1, anti-Rab7, anti-LAMP2, and anti-Cx43 antibodies were added overnight, cells were stained with fluorescein-labeled secondary antibodies for 2 h. Subsequently, cells were incubated with DAPI to stain nuclei, then the slide was completely sealed with nail polish. Images were acquired using a Leica inverted laser confocal microscope (Wetzlar, Germany).

### 4.4. Transmission Electron Microscopy 

For electron microscopy, BRL 3A cells were seeded in 6-well plates. When hepatocytes grew to 50% confluence, they were incubated with either 0 μM or 10 μM Cd and cultured in an incubator for 12 h. DMEM was discarded, and 1 mL of trypsin was added to each well. After 30 s, adherent cells were detached and collected into 1.5 mL tubes. Hepatocytes were then centrifuged at 1000 r/min in a low-speed centrifuge, the supernatant was discarded, and the precipitate was fixed with an electron microscope fixative. An additional sample was prepared to observe cell morphology under an electron microscope. Images were observed using a transmission electron microscope (HT7800, HITACHI, Japan).

### 4.5. Western Blot Assay

Cells were scraped and lysed with 50 mmol/L HEPES buffer containing 0.5% nonidet P-40, proteinase, and phosphatase inhibitors (1 mM phenylmethylsulfonyl fluoride, 1 mg/mL each of aprotinin, leupeptin, and pepstatin, and 1 mmol/L Na_3_VO_4_, 25 mmol/L NaF, respectively). The cellular lysate was sonicated and centrifuged at 11,000× *g* for 10 min at 4 °C. Protein concentration was determined using the Bradford assay. Proteins (20 mg) were separated via SDS-PAGE and transferred onto polyvinylidene fluoride (PVDF) membranes. These blots were blocked with 5% skimmed milk or BSA (bovine serum albumin) (for P-Cx43) in Tris-buffered saline containing 0.1% Tween 20 (TBS-T) for 2 h and incubated overnight at 4 °C with a primary antibody (each antibody used for Western blotting was diluted to 1:1000, the anti-GAPDH antibody was diluted to 1:5000). Blots were then washed three times with TBS-T and incubated for 2 h with horseradish peroxidase-conjugated secondary antibody in TBS-T. After the blots were washed three times, they were visualized using enhanced chemiluminescence reagents.

### 4.6. Statistical Analysis 

The results were expressed as the mean ± standard deviation (SD). All Western blotting results were representative of at least three experiments. Statistical analysis was assessed by the student’s unpaired *t*-test for the study that had 2 groups. Statistical analyses were performed using one-way analysis of variance (ANOVA) for studies with 4 groups, followed by Tukey’s test. All statistical analyses were performed using GraphPad software package (Prism 8.0) for Windows. Differences were considered statistically significant at *p*-values of less than 0.05.

## 5. Conclusions

Our study shows that Cd can induce the production of AGJs in hepatocytes, and these AGJs can be degraded by endosome–lysosomes, which is a temporary protective response of liver cells to Cd damage (Figure 6). Our results suggest the possibility of AGJs induced by other environmental poisons or harmful stimuli: as long as the harmful stimulus is sufficiently effective, AGJs are produced between cells to rapidly reduce intercellular communication and contact to mitigate damage. In addition, our study suggests that researchers should pay attention to the phenomenon of AGJs production in different kinds of cells and tissues under different harmful stimuli. Especially in the study of exosomes, it is necessary for scholars to deeply explore how the AGJs secreted into the extracellular cells affect the function between tissues and cells.

## Figures and Tables

**Figure 1 ijms-23-15607-f001:**
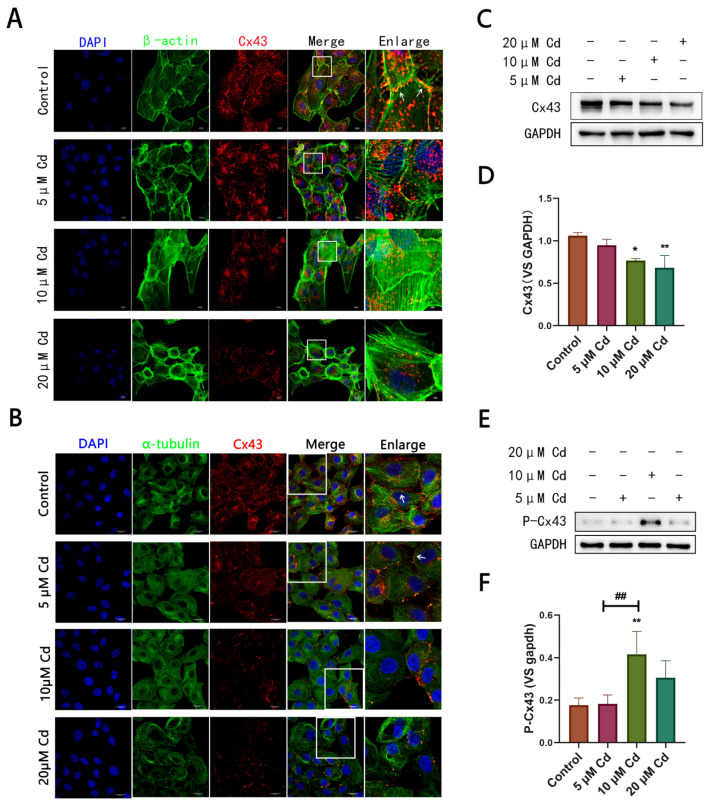
Cd induces Cx43 internalization and degradation in hepatocytes: (**A**) Cx43 was co-stained with microfilaments (labeled with phalloidin). Green represents microfilaments, and red represents Cx43. The white rectangle represents Cx43 between cells under different Cd concentrations. The white arrow represents GJP. Scale bar = 10 μm; (**B**) Anti-Cx43 antibody was co-stained with anti-α-tubulin-FITC antibody. Green represents maicrotubules, and red represents Cx43. The white rectangle represents Cx43 between cells under different Cd concentrations. The white arrow represents GJP. Scale bar = 10 μm; (**C**,**D**) Western blotting analysis of the Cx43 level in rat hepatocytes. Statistical results were obtained from at least three independent experiments; * *p* < 0.05 compared to the control group, ** *p* < 0.01 compared to the control group; (**E**,**F**) Western blotting analysis of P-Cx43 (Ser 373) level; ** *p* < 0.01 compared to the control group, ## *p* < 0.01 compared to the Cd group.

**Figure 2 ijms-23-15607-f002:**
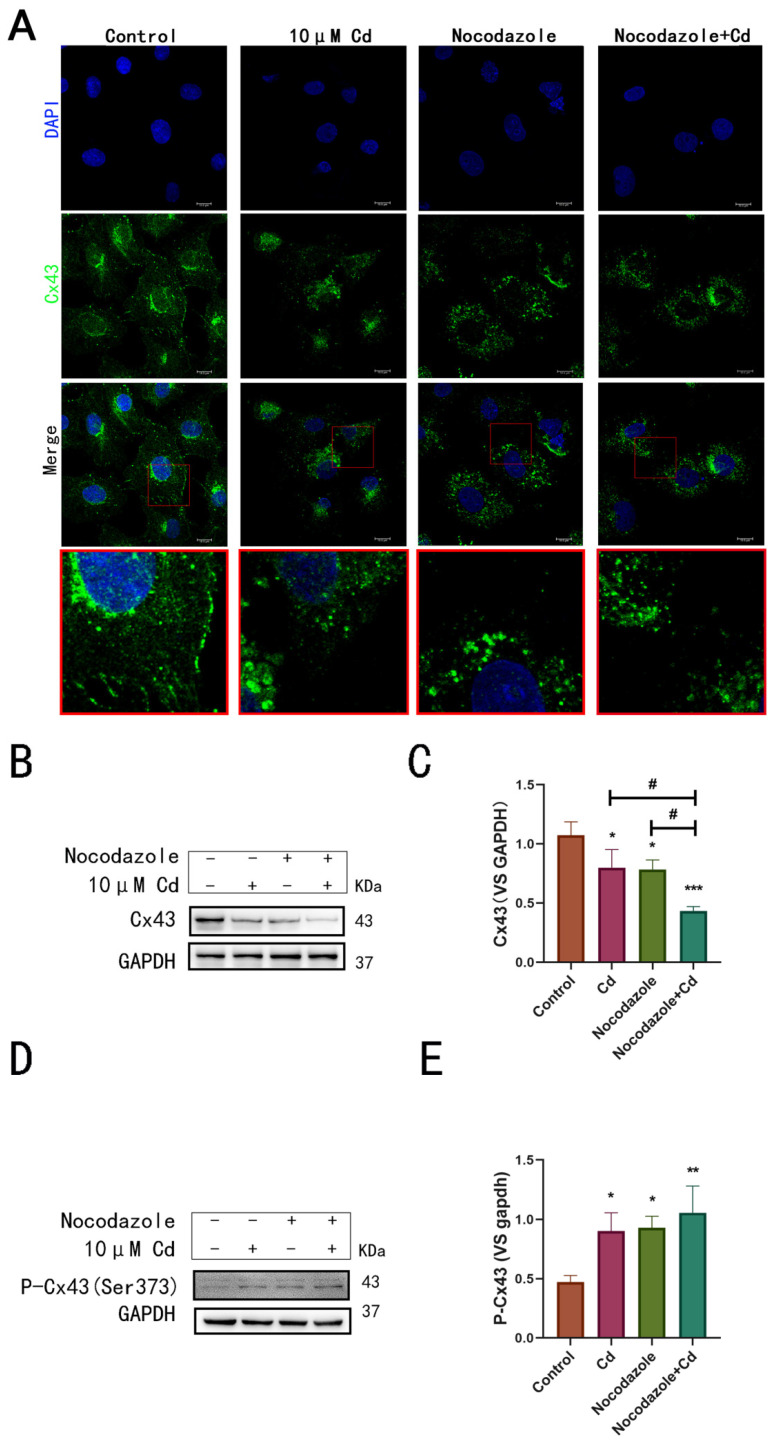
Microtubule damage from Cd toxicity promotes the internalization of Cx43 in hepatocytes: (**A**) Immunofluorescence intracellular distribution of Cx43 after the combined administration of Cd and the microtubule inhibitor nocodazole. Scale bar = 10 μm; (**B**,**C**) Analysis of Cx43 level after combined treatment with Cd and nocodazole using Western blotting. * *p* < 0.05 compared to the control group; *** *p* < 0.001 compared to the control group; # *p* < 0.05 compared to the Cd group; (**D**,**E**) Analysis of P-Cx43 (Ser373) levels after combined treatment with Cd and nocodazole using immunoblotting. * *p* < 0.05 compared to the control group; ** *p* < 0.01 compared to the control group.

**Figure 3 ijms-23-15607-f003:**
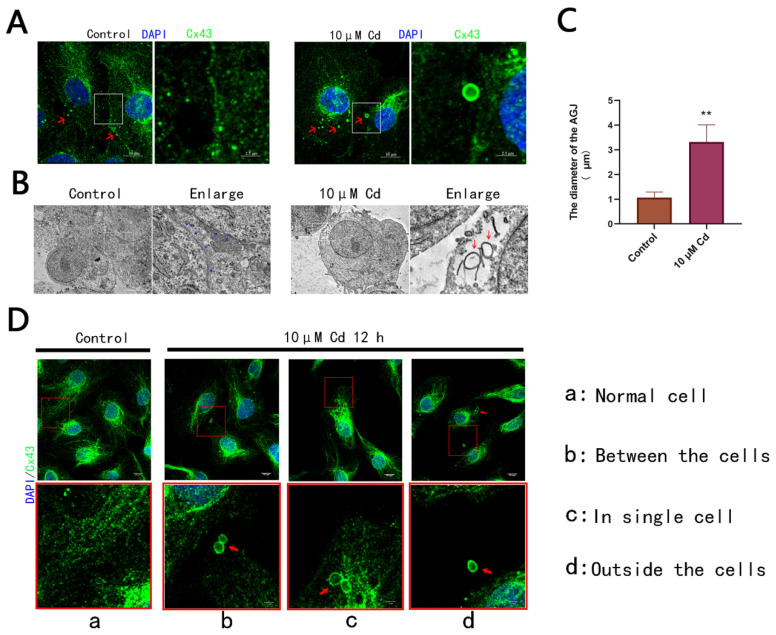
Cd induces the formation of AGJ in rat hepatocytes: (**A**) Cd exposure induced the formation of AGJs between cells in rat hepatocytes. Red arrows represent Cx43 spots or AGJs. Scale bar = 10 μm; (**B**) AGJs observed under a transmission electron microscope. Blue arrows represent cell junction regions, and red arrows represent the AGJ of the bilayer membrane. Scale bar = 5 μm; (**C**) Image J software was used to determine the diameters of AGJ and normal Cx43 spots and eventually calculate the ratio. This method is a way to manually count the particle diameter in images provided by Image J software: set a ruler for images, and then manually measure the diameter of Cx43 fluorescent sites (13 in total, 1 fluorescent site per cell) in cells of the control group and the diameter of all AGJ (13 in total) in cells of Cd group, and then use GraphPad Prism software to compare the difference between two groups of data. The results were derived from measuring the diameters of 13 different AGJs. ** *p* < 0.01 compared to the control group; (**D**) Cd exposure induced the formation of AGJs in different regions of rat hepatocytes. Red arrows represent the structure of AGJ. Scale bar = 10 μm.

**Figure 4 ijms-23-15607-f004:**
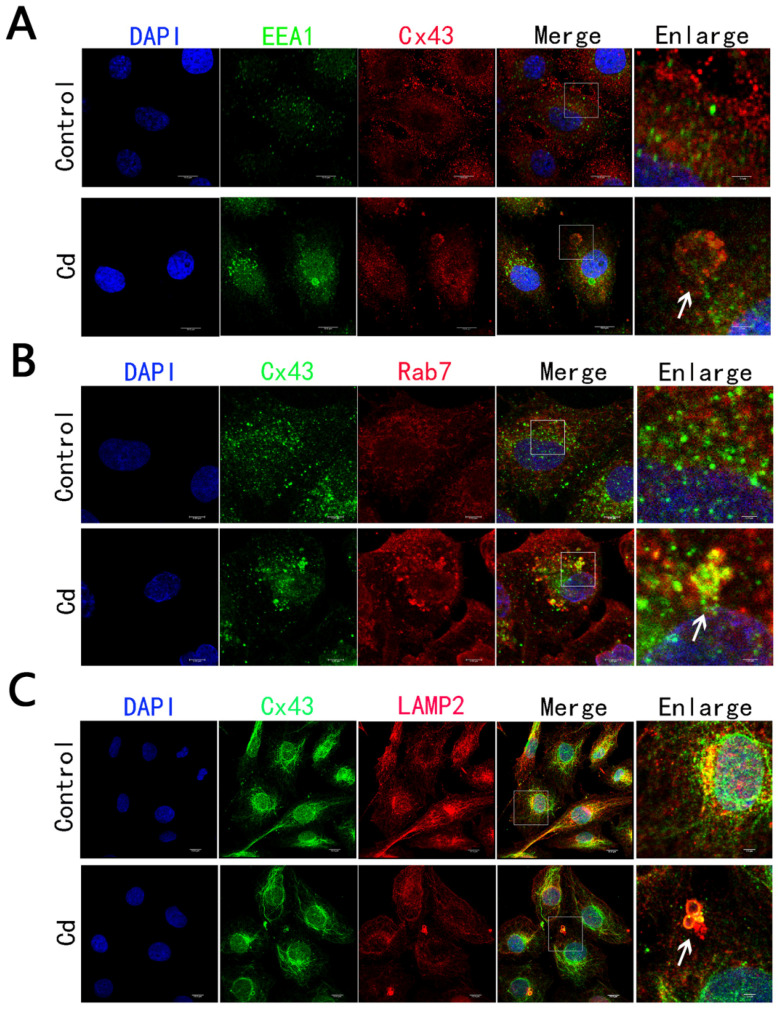
Cx43 interacted with endosomes/lysosomes after internalization: (**A**) Anti-Cx43 antibody (red) is co-stained with antibody against early endosomal marker EEA1 (green), white arrow represents AGJ. Scale bar = 10 μm; (**B**) Anti-Cx43 antibody (green) is co-stained with antibody against late endosomal marker RAB7a, white arrow represents AGJ. Scale bar = 10 μm; (**C**) Antibody against Cx43 (green) were co-stained with the anti-lysosomal marker protein LAMP2, AGJ was represented by the white arrow. Scale bar = 10 μm.

**Figure 5 ijms-23-15607-f005:**
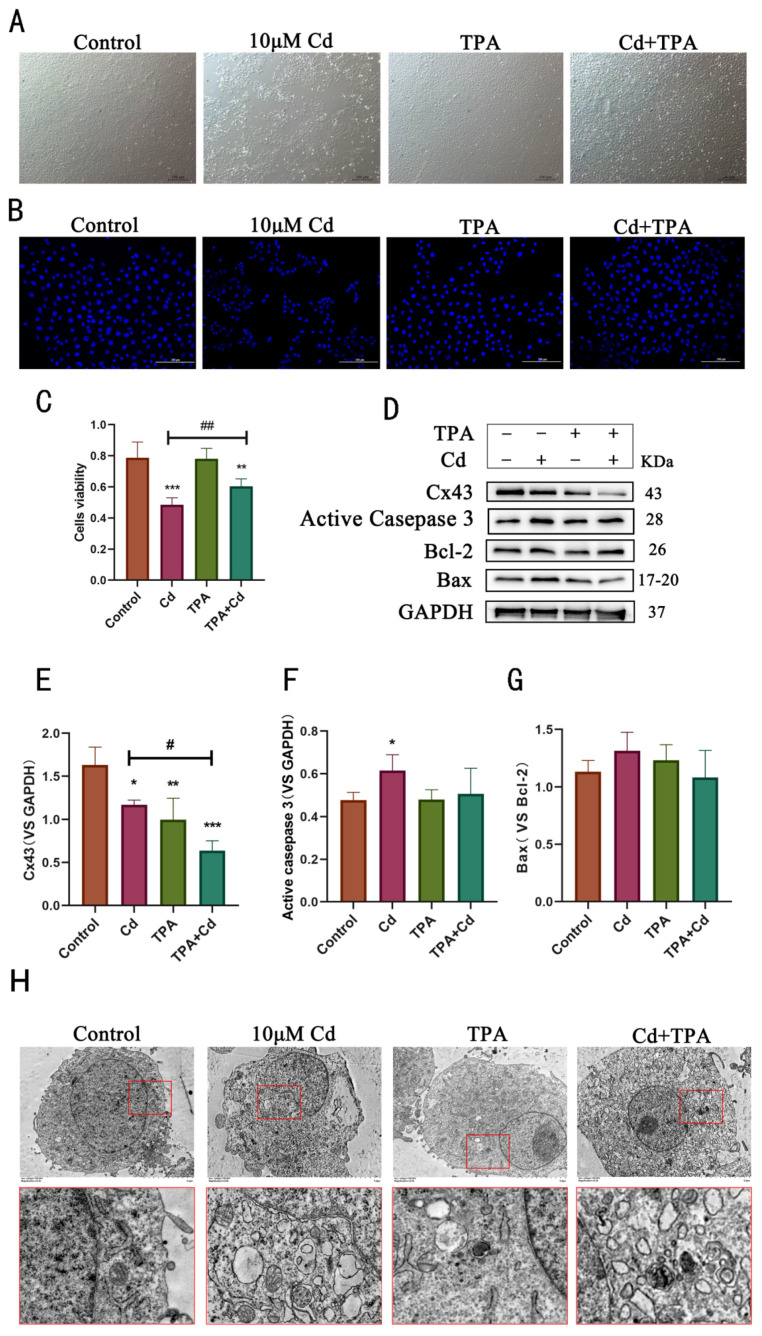
Application of TPA alleviated cell damage caused by Cd exposure in rat hepatocytes: (**A**) Light microscope observation revealed that co-treatment with TPA and Cd reduces cell death caused by Cd. Scale bar = 100 μm; (**B**) TPA alleviates Cd-induced damage to the rat cell nuclei. Scale bar = 200 μm; (**C**) CCK 8 assay results showing TPA alleviates Cd-induced reduction in the viability of rat hepatocytes, ** *p* < 0.01, vs. control; *** *p* < 0.001, vs. control; ## *p* < 0.01, vs. Cd; (**D**–**G**) Western blotting analysis of the levels of Cx43 and apoptotic proteins (activated-caspase 3, ratio of Bax to Bcl-2); * *p* < 0.05 compared to the control group; ** *p* < 0.01 compared to the control group; *** *p* < 0.001 compared to the control group; # *p* < 0.05 compared to the Cd group; (**H**) Observation of cell damage in rats treated with Cd and TPA by transmission electron microscope. Scale bar = 5 μm.

**Figure 6 ijms-23-15607-f006:**
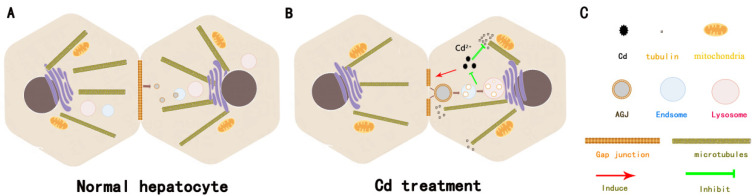
Alleviation of Cd-induced toxicity in rat hepatocytes through the formation of AGJs and degradation via endosome–lysosome pathway: (**A**) Small-scale degradation of Cx43 in normal cells; (**B**) Internalization of Cx43 into large AGJs by hepatocytes. The AGJs can fuse with early endosomes, develop into late endosomes, and finally fuse with lysosomes. Internalization of Cx43 attenuates Cd-induced cell damage and apoptosis through an unknown pathway. (**C**) Explanation of symbols and patterns in Figure (**A**) and (**B**).

## Data Availability

The data that support the findings of this study are available from the corresponding author upon reasonable request.

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
