# Peer review of "Rat Hepatocytes Mitigate Cadmium Toxicity by Forming Annular Gap Junctions and Degrading Them via Endosome–Lysosome Pathway"

_ijms, 2022, doi:10.3390/ijms232415607_

Round 1

Reviewer 1 Report

Rat hepatocytes mitigate cadmium toxicity by forming annular gap junctions and degrading them via endosome-lysosome pathway. It is merit, however, the following should be paid attention to:

What was the selection criteria of cadmium and nocodazole dose?

In Fig. 2C, the author wrote ***, but it is not present in figure caption.

Reviewer 2 Report

1. Many items of cited reference (e.i. 6, 11, 14, 15, 16, 22, 24, 25, 34, 36, 37, 38, 40) should be exchanged for articles published in recent years. Some of the articles cited (e.i. 40) were published over 20 years ago, which is unacceptable in this area of knowledge

2. Conclusions should be rewritten, because those that are in the manuscript are rather observations or a repeat description of the results

Reviewer 3 Report

Dear Editor and Authors

The manuscript presents an interesting approach in advancing knowledge about Cd dynamics in the cellular environment. The manuscript is well written and appears to comply with quality control protocols for experiments and analyzes. I consider that the study should be published after a minor review, considering the topics below.

Abstract: The text must introduce briefly, what are Connexin 43, Ser373 and must be written in full AGJ and BRL.

Line 28: Cd is not an inert element, as it induces toxicological effects in organisms. Please review the sentence.

Lines 29-30: Aquatic or terrestrial organisms may be at risk, not water and soil. Please, review the sentence.

Round 2

Reviewer 2 Report

Thank you very much

Author Response

Thank you very much!